



# 1 Variability and Trends in Physical and Biogeochemical
# 2 Parameters of the Mediterranean Sea during a Cruise with
# 3 RV MARIA S. MERIAN in March 2018

**Dagmar Hainbucher[1], Marta Álvarez[4], Blanca Astray Uceda[4], Giancarlo Bachi[5],**
**Vanessa Cardin[3], Paolo Celentano[6], Spyros Chaikakis[7], Maria del Mar Chaves**
**Montero[3,9], Giuseppe Civitarese[3], Noelia M. Fajar [4], Francois Fripiat[10], Lennart**
**Gerke[2], Alexandra Gogou[7], Elisa Fernández Guallart[4], Birte Gülk[1], Abed El**
**Rahman Hassoun[8], Nico Lange[2], Andrea Rochner[1], Chiara Santinelli[5], Tobias**
**Steinhoff[2], Toste Tanhua[2], Lidia Urbini[3], Dimitrios Velaoras[7], Fabian Wolf[2],**
**Andreas Welsch[1]**
[1]{Institut für Meereskunde, CEN, Universität Hamburg, Bundesstraße 53, 20146 Hamburg,
Germany}
[2]{GEOMAR, Helmholtz-Zentrum für Ozeanforschung Kiel, Wischhofstr. 1-3, 24148 Kiel,
Germany }
[3]{Istituto Nazionale di Oceanografia e di Geofisica Sperimentale – OGS, Dept. Of
Oceanography, Borgo Grotta Gigante 42/c, 34010 Sgonico, Trieste, Italy}
[4]{Instituto Español de Oceanografía (IEO), centro de A Coruña, Spain}
[5]{ Istituto di Biofisica, CNR, Pisa, Italy}
[6]{Istituto di Scienze Marine, Venezia, Italy}
[7]{Helenic Centre for Marine Research, Athens, Greek}
[8]{National Council for Scientific Research in Lebanon. National Center for Marine Sciences}
[9]{ Centro Euro-Mediterraneo sui Cambiamenti Climatici CMCC, Bologna, Italy}
[10]{Max Planck Institute for Chemistry, Mainz, Germany}
*Correspondence to:* Dagmar Hainbucher (dagmar.hainbucher@uni-hamburg.de)

## 27 Abstract

The last decades have seen dramatic changes in the hydrography and biogeochemistry of the
Mediterranean Sea. The complex bathymetry, highly variable spatial and temporal scales of
atmospheric forcing and internal processes contribute to generate complex and unsteady
circulation patterns and significant variability in biogeochemical systems. Part of this



variability can be influenced by anthropogenic contributions. Consequently, it is necessary to
document details and to understand trends in place to better relate the observed processes and
to possibly predict the consequences of these changes. In this context we report on data from
an oceanographic cruise in the Mediterranean Sea on the German research vessel MARIA S.
MERIAN (MSM72) in March 2018. The main objective of the cruise was to contribute to the
understanding of long-term changes and trends in physical and biogeochemical parameters,
such as the anthropogenic carbon uptake and to further assess the hydrographical situation after
the major climatological shifts in the eastern and western part of the basin, known as the Eastern
and Western Mediterranean Transients. During the cruise, multidisciplinary measurements
were conducted on a predominantly zonal section throughout the Mediterranean Sea,
contributing to the global GO-SHIP repeating hydrography program and adhering to the GO-
SHIP requirements.
**Data coverage and parameter measured**
Repository-Reference (table 1a and table 1b):
Table 1a. List of physical parameters from Maria S. Merian cruise MSM72 as seen in the
PANGAEA database. PI: Dagmar Hainbucher

| Parameter Name | Short name | Unit | Method | Comments |
|---|---|---|---|---|
| DATE/TIME | Date/Time | | | Geocode |
| LATITUDE | Latitude | | | Geocode |
| LONGITUDE | Longitude | | | Geocode |
| Pressure, water | Press | dbar | CTD, SEA_BIRD SBE 911plus | |
| Temperature, water | Temp | °C | CTD, SEA_BIRD SBE 911plus | |
| Salinity | Sal | | CTD, SEA_BIRD SBE 911plus | PSU |
| Oxygen | O2 | µmol/l | CTD with attached oxygen sensor calibrated, corrected using Winkler titration | |
| Pressure, water | Press | dbar | UnderwayCTD (UCTD), Oceanscience | |
| Temperature, water | Temp | °C | UnderwayCTD (UCTD), Oceanscience | |
| Salinity | Sal | | UnderwayCTD (UCTD), Oceanscience | PSU |
| DEPTH, water | Depth | m | | |





| Current velocity east-west | UC | m/s | Shipboard Acoustic Doppler Current Profiling (SADCP) |
| Current velocity north-south | VC | m/s | Shipboard Acoustic Doppler Current Profiling (SADCP) |
| DEPTH, water | Depth | m | |
| Current velocity east-west | UC | m/s | lowered Acoustic Doppler Current Profiling (lADCP) |
| Current velocity north-south | VC | m/s | lowered Acoustic Doppler Current Profiling (lADCP) |

2    Table 1b. List of biogeochemical parameters from Maria S. Merian cruise MSM72 as seen in

3    the CCHDO database. PI: Toste Tanhua

| Variable | Unit |
| --- | --- |
| Dissolved Oxygen ($O_2$) | µmol kg$^{-1}$ |
| Sulphurhexafluorid ($SF_6$) | fmol kg$^{-1}$ |
| $CCl_2F_2$ (CFC-12) | pmol kg$^{-1}$ |
| Nitrate ($NO_3^-$) | µmol kg$^{-1}$ |
| Nitrite ($NO_2^-$) | µmol kg$^{-1}$ |
| Phosphate ($PO_4^{2-}$) | µmol kg$^{-1}$ |
| Silicate (Si) | µmol kg$^{-1}$ |
| Dissolved Inorganic Carbon (DIC) | µmol kg$^{-1}$ |
| Total Alkalinity (TA) | µmol kg$^{-1}$ |
| pH | Total scale @ 25°C |
| Carbonate ($CO_3^{2-}$) | µmol kg$^{-1}$ |
| $\delta^{13}C$ of DIC | Per mille |
| Total Dissolved Nitrogen (TDN) | µmol kg$^{-1}$ |
| Total Dissolve Phosphorus (TDP) | µmol kg$^{-1}$ |
| $CHClF_2$ (HCFC-22) | pmol kg$^{-1}$ |



| | |
|---|---|
| $C_2H_3Cl_2F$ (HCFC-141b) | pmol kg$^{-1}$ |
| $C_2H_3ClF_2$ (HCFC-142b) | pmol kg$^{-1}$ |
| $CH_2FCF_3$ (HFC-134a) | pmol kg$^{-1}$ |
| $C_2HF_5$ (HFC-125) | pmol kg$^{-1}$ |
| $CHF_3$ (HFC-23) | pmol kg$^{-1}$ |

https://doi.pangaea.de/10.1594/PANGAEA.905902          (for CTD)
https://doi.pangaea.de/10.1594/PANGAEA.913512          (for UCTD)
https://doi.pangaea.de/10.1594/PANGAEA.913608          (for ADCP)
https://doi.pangaea.de/10.1594/PANGAEA.913505          (for lADCP)
https://doi.pangaea.de/10.1594/PANGAEA.905887      (for chemical data)
https://doi.org/10.25921/z7en-hn85                       (for pCO$_2$)
A link to the summary page of the cruise MSM72 can be found in the PANGAEA data base
under: https://www.pangaea.de/?q=msm72&f.campaign%5B%5D=MSM72
Coverage: 34°N-41°N, 6°W-28°E
Location Name: The Mediterranean Sea
Date/Time Start: 2. March 2018
Date/Time End: 3. April 2018
**1.      Introduction**
Contrary to earlier ideas that the Mediterranean Sea is always in a steady state, we now know
in the light of new research that the Mediterranean Sea is not but it is potentially sensitive to
climatic changes (Malanotte-Rizzoli, 2014). Proof of this are the drastic changes that the eastern
Mediterranean (EMed) has undergone in the past. The largest climatic event, named Eastern
Mediterranean Transient (EMT), occurred in the EMed between the late 1980's and early
1990's, where deep-water formation switched from the Adriatic to the Aegean Sea. This
episode modified the thermohaline characteristics of the outflow through the Sicily Channel,
changing the characteristics of the western Mediterranean (WMed) accordingly  (Millot et al.,



2006, Schroeder et al., 2006). Thus, since 2005, the deep waters of the WMed have undergone
significant physical changes, which are comparable to the EMT, both in terms of intensity and
observed effects (Schroeder et al., 2008). This event is called the Western Mediterranean
Transient (WMT). The existence of both transients contradicts the hypothesis of a steady state.
On the other hand, it has also been proven that an EMT has never been observed before (Roether
et al., 2013).
The characteristic of the Mediterranean Sea is also such that it has the potential to sequester
large amounts of anthropogenic $CO_2$, Cant, since the Mediterranean Sea has high alkalinity and
temperature, which can be rapidly transported to deep by the overturning circulation (e.g.
Schneider et al., 2010). The column inventories of Cant in the Mediterranean are among the
highest found in the world oceans; the Mediterranean Sea thus stores a significant portion of
the global anthropogenic emissions of Cant despite its relatively small volume.
Furthermore, marine dissolved organic carbon (DOC) represents the largest reservoir of
reduced carbon ($662 \cdot 10^{15}$ g C) on Earth (Hansell, 2009), it therefore plays a major role in the
global carbon cycle. Its role in the functioning of marine ecosystems is equally crucial since
DOC is released at all the levels of the food web, as a byproduct of many trophic interactions
and/or metabolic processes and is the main source of energy for the heterotrophic prokaryotes
(Carlson and Hansell, 2015). Although most of DOC is produced in-situ, external sources
(atmosphere, rivers, sediments) may affect its concentration and distribution. Physical
processes, such as deep-water formation, thermohaline circulation, vertical stratification and
mesoscale activities have been reported to be the main drivers of DOC distribution in the
Mediterranean Sea (Santinelli, 2015, Santinelli et al., 2015, Santinelli et al., 2013, Santinelli,

23   2010).

The main scientific objective of the cruise reported here was to add knowledge to the different
scales and magnitudes of variability and trends in circulation, hydrography, and
biogeochemistry of the Mediterranean Sea. Key variables were measured in strategic regions
in order to understand changes, the reason for occurrence, and the drivers.
The following science questions were addressed:
1.     What are the long-term changes and/or trends in physics and biochemistry in the
Mediterranean Sea, including all the sub-basins?
2.     How is the hydrographic situation in the Mediterranean developing further after the EMT
and WMT? Is there still a tendency of the system to return to the pre-EMT situation and is there



a similar trend in the WMed?
3.    How are eddies distributed in the EMed and WMed during the cruise? Do they differ in
the sub basins? To what extent is heat and salt transferred into the vertical by eddies in the
WMed and EMed during the cruise period?
4.    What is the uptake rate of the anthropogenic carbon in the Mediterranean and is this
changing over time?
5.    What is the extent of the variability and trends in the inventory of biogeochemical
variables (including oxygen, nutrients and DOC)?
6.    What are the baseline values of rarely measured Essential Ocean Variables (EOVs) such
as dissolved organic carbon and nitrous oxide?

## 2.    Data Provenance

The survey was carried out on the German RV Maria S. MERIAN from 2$^{nd}$ of March to 3$^{rd}$ of
April 2018. The cruise started on Iraklion, Greece and ended in Cadiz, Spain. The main focus
of the cruise was on an east-west transect across the Western and Eastern Mediterranean Sea
(figure 1) starting east of Crete and ending near the Strait of Gibraltar, which is a repeating
hydrographic line in GO-SHIP (MED1). Difficulties with diplomatic authorizations for Marine
Scientific Research (MSR) in the disputed EEZ between Greek and Turkey made it impossible
for us to carry out our measurements in this area, so that no data were obtained east of Kasos
Strait.


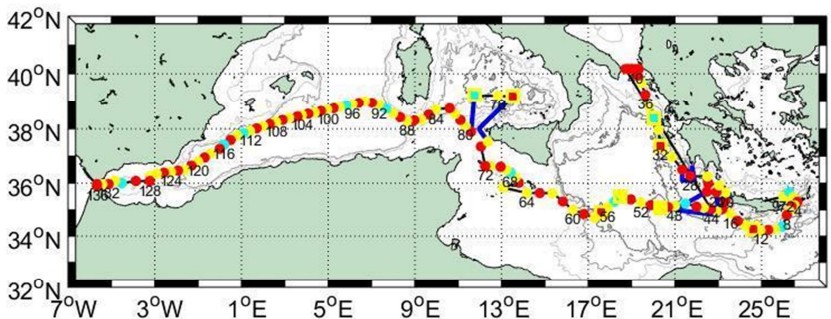

Figure 1: Station Map. Yellow dots: CTD without any chemical sampling, red dots: CTD with
chemical sampling, cyan dots: CTD with chemical and additional sampling of isotopes, yellow
squares: deployment of drifter and floats, blue lines: fine resolved uCTD and ADCP tracks.
Black lines: Track with uCTD casts between CTD stations.
During the thirty-three days of the cruise we carried out measurements of hydrographic and
biogeochemical variables along-track with the classical approach i.e. CTD, lADCP, uCTD
instrumentation and bottle samples on highly resolved sections across the Mediterranean Sea.
The high resolution of CTD stations, enhanced for the physical parameters by additional uCTD
measurements, allowed us to resolve the eddy field on the sections, the analysis was also
supported and complemented by satellite data.
Most sections and CTD-positions follow previous sampling strategies (cruise M84 and other
along the GO-SHIP line MED-01, i.e. Tanhua et al., 2013) to allow long-term trend analyses.
Along the different sections, CTD stations including sampling of chemical parameters were
conducted approximately every 30 nm, CTD without sampling about every 15-20 nm and with
even smaller spacing in the Straits. In addition, underway CTD measurements and ADCP
measurements were performed between CTD stations.
The water sampling program included measurements of all level 1 variables as defined by GO-
SHIP (i.e. oxygen, macronutrients, transient tracers and the carbonate system, http://www.go-
ship.org/DatReq.html) and measurements of the biogeochemical EOVs $^{13}$C, nitrous oxide
($N_2O$) and dissolved organic carbon (DOC). These data were used to quantify trends and



variability of ventilation and biogeochemical cycles, in particular uptake of anthropogenic
carbon.
Sections were additionally conducted through the important passages: the Strait of Otranto,
Kasos Strait, Antikythera Strait, Strait of Sicily and Strait of Gibraltar, in order to characterize
the incoming and outgoing flows. CTD stations in the Eastern Ionian Sea were carried out to
quantify the flow of the Levantine Surface Water (LSW) into the Adriatic Sea and to track the
outflow of the Adriatic Deep Water (AdDW) into the Ionian.

## 9   3.   Methods

### 10   3.1 CTD/rosette

Altogether 136 CTD cast were performed from which 18 catalogued as isotopic (a full suite of
observations), 65 as chemical (i.e. all GO-SHIP level 1 variables), and 59 as physical (i.e. only
sampling for salinity). Due to the water amount needed, 2 casts were performed on most of the
isotopic stations, the first cast was a full profile and the second a shallow one. During the
physical stations water samples at 3 levels were taken for salinity analysis. The samples were
then analyzed on board using a Guildline Autosal Salinometer. A total of 162 samples in 59
stations were taken during the cruise with an offset with respect to standard water varying from
0.0002 to 0.0030 depending on the laboratory temperature.
The primary CTD system (specifications see table 2) initially used on board was a Seabird
SBE9plus + CTD s/n 0285 from the University of Hamburg connected to a SBE11 deck unit,
configured with a 24- position SBE-32 pylon (from GEOMAR) with 10-liter Niskin bottles.
Position of bottles #23 and #24 was occupied by the lADCP (specifications see table 3).
Initially, the CTD was set up with two sensors for temperature and conductivity, an oxygen
sensor, a fluorometer and an altimeter. To test the configuration and performance of the
instrument a station was carried out on the Cretan Sea at the start of the cruise. Unfortunately,
we had countless problems with instruments, sensors, cables and rosette during most of the
campaign which forced us to change them very often with others available on board resulting
in a continuous change of system configuration. Thus, all different configurations were
carefully considered when post-processing the CTD data.
Temperature, salinity and pressure data were post-processed by applying Seabird software and
MATLAB routines. At this stage, spikes were removed, 1 dbar averages calculated. A first
attempt to assess the performance of the conductivity sensors installed on the CTD-Rosette was





done by comparing the salinity data with the bottle samples analyzed with the salinometer. The
different hardware setups and configurations are taken carefully into account during post-
processing. Overall accuracies are within the expected range of salinity (0.003).
Table 2: Used CTD instrument and sensors. Owner of instruments are either the University of
Hamburg, Germany (IfM-HH), the National Institute of Oceanography and Geophysics (OGS),
Italy or the property of the vessel MERIAN (MSM).

| Instrument/Sensor | Serial Number (owner) |
|---|---|
| SBE 911plus / 917plus CTD | 285 (IfM-HH) <br> 806 (MSM) <br> 807 (MSM) |
| Temperature 1: SBE-3-02/F | 1717 (OGS) <br> 5716 (MSM) |
| Conductivity 1: SBE-4-02/2 | 3442 (OGS) <br> 4152 (MSM) |
| Temperature 2: SBE-3-02/F | 1294 (IfM-HH) <br> 5719 (MSM) |
| Conductivity 2: SBE-4-02/2 | 1106 (IfM-HH) <br> 4159 (MSM) |
| Oxygen 1 SBE 43 | 3392 (OGS) <br> 2417 (MSM) <br> 0951 (MSM) |
| Oxygen 2 SBE 43 | 1761 (IfM-HH) <br> 2418 (MSM) <br> 0881 (MSM) |
| Fluorometer WETLAB <br><br> SeaPoint | 1755 (MSM) <br> 1754 (MSM) <br> SCF2874 |

**3.2 Underway-CTD**
Underway CTDs measurements (uCTD, specifications see table 4) provide high-resolution
profiles of temperature, conductivity and depth, which allow to characterize the upper ocean
properties and to identify the position and characteristics of mesoscale structures. The
advantage of this type of measurements is that it is not required to stop the vessel, but only to



maintain lower velocities (about 3 kn) during the deployments  to reach greater depths. These
measurements were made with an Ocean Science uCTD system.
The first uCTD deployment was done on March 5[th], between CTD 015 and 016 stations, and
we continued with this type of sampling between each CTD station to increase the sampling
resolution. Unfortunately, several deployments were cancelled due to severe weather conditions
and  no uCTD cast was performed when the depth was shallower than 500m. Altogether 176
casts were taken with depths ranging from 557 to 864 m.
Two probes were used during the cruise with a no time limit mode configuration (apart from
the first cast configured to stop recording after 600 seconds, reaching 616 m depth) in order to
get longer records. The probe tail spools were attached to the winch through a rope loop that
was made new every day in the morning. Despite the probes can record several casts, data were
downloaded right after each cast using a SBE software in order to avoid losing the data in case
the probe was lost, and to free the memory. The probes were exchanged when the battery was
running low (around 3.8V). In three occasions, no data were recorded because the magnet was
taken off twice before deployment.
For calibration purposes, some additional casts were done right after the CTD cast in order to
compare the data sets. The probes were also sent down with the starboard CTD in station 130.
Data files were processed using a set of Matlab routines. After extracting the downcast data, a
first correction was done for removing inaccuracies in the descend rate based on the work of
Ullmann and Hebert (2013). Additionally, the data were aligned to the comparable CTD data
sets.
Table 3: Used UCTD sensors.

| Probe 1 | Device Type | Serial Number (owner) |
|---------|-------------|------------------------|
| 0289 | 90745 uCTD /SBE49 FastCat CTD | 702-0289 (IfM-HH) |
| 0183 | 90745 uCTD /SBE 49 FastCat CTD | 702-0183 (IfM-HH) |

**3.3 lADCP Measurements**
The ocean current was studied by means of vertical profiles made with a lADCP-2 system
(Workhorse RD Instruments type, table 3) which included two ADCPs operating at a frequency
of 300 kHz, one looking upward and the other one looking downward. The system was placed
in the rosette occupying the position of Niskin bottles 23 and 24. During the cruise, the lADCP
batteries were changed twice: the first time on March 17[th] in Station 58 and the second time on



March 27th in Station 105. LADCP measurements were done at all CTD stations except for
three (Station 73, 74, 80) with water depth less than 500 m. For these stations, the currents were
observed by the ship mounted ADCP. At double (isotope) stations, lADCP profiles were only
recorded from the deep cast. The gained data were processed with LDEO Matlab LADCP-
processing system Version 10.15 (Turnherr, 2014). This software uses the raw lADCP data,
processed CTD data and navigational data from the CTD. The resulting data are the u- and v-
velocities at the depth. The bin size was set to 8m.
Table 4: Used lADCP.

| Device Type | Serial Number (owner) |
|---|---|
| WHM300 | Master s/n #22762 (IfM-HH) |
| WHM300 | Slave s/n #22763 (IfM-HH) |
| | |

**11     Shipborne ADCP**

During the whole campaign, underway current measurements were taken with two vessel-
mounted VM-ADCPs Ocean Surveyor (ADCP) manufactured by RDI. The first, with work
frequency of 75 kHz, covered approximately the top 500-700m of the water column. The
number of bins was set to 100 with bin size of 8 m. The second, with work frequency of 38
kHz, has a depth range of about 1600 m, set with the same bin number as the previous one and
bin size of 16 m. Both instruments run in narrowband mode and were controlled by computers
using the conventional RDI VMDAS software under a MS Windows system with a pinging set
to fast as possible. No interferences with other used acoustical instruments were observed. The
ADCP data was afterwards post-processed with the CODAS3 Software System
(https://currents.soest.hawaii.edu/docs/adcp_doc/), which allows extracting data, assigning
coordinates, editing and correcting velocity data. Moreover, the data were corrected for errors
in the value of sound velocity in water, and misalignment of the instrument with respect to the
axis of the ship (about -2.8 degrees for 75 kHz ADCP and about -0.15 degrees for 38 kHz
ADCP).



**3.5 Underway CO$_2$ and O$_2$ Measurements**

Underway (UW) measurements of partial pressure of CO$_2$, and dissolved oxygen concentrations in seawater were carried out by means of a Contros HydroC pCO$_2$ analyzer for pCO$_2$ and an Aanderaa optode for oxygen.

The instruments were placed in a cooling box in the hangar. Seawater was drawn from the ship's centrifugal pump for clean seawater that was continuously flowing through the cooling box with the inlet close to the instruments. Water was pumped through a SeaBird 5 salinity and temperature sensor and on to the HydroC instrument (Gerke et al., 2020).

The system operated reliably throughout the cruise, except when data acquisition was interrupted for the pCO$_2$ instrument for 2 days directly after the ship's centrifugal pump was switched off. This led to a gap 5-days period without data between March 5$^{th}$ and 10$^{th}$. During the cruise 13 samples were taken from the cooling box for discrete measurements of pH and total alkalinity. The UW measurements started on March 2$^{nd}$ at 20:20 and stopped on April 1$^{st}$, 2018, at 14:00 (UTC).

The underway oxygen measurements were calibrated by comparing to the Winkler measurements taken for surface samples at the CTD stations

**3.6 Dissolved Oxygen**

Dissolved oxygen in seawater was measured at every station and depth along the cruise and reported in μmol/kg. Oxygen was measured following the automatic Winkler potentiometric method modified after Langdon (2010). Titrations were done within the sampling calibrated flasks using an Automatic Titrator Mettler Toledo T50 with a platinum combined electrode.

Reagents blank and Thiosulphate standardization were done daily by means of Potassium Iodate Standard 1.667 millimolar by OSIL, UK. About 1400 samples were analyzed on board. The precision of dissolved oxygen measurements was determined on five replicates, at the beginning and at the end of the cruise (table 5).

In addition, during the cruise 46 duplicates were analysed. The results are given in table 6.



Table 5: Precision of dissolved oxygen. (STD = standard deviation, CV = Coefficient of
Variation)

| Parameter | Beginning of the cruise | | | End of the cruise | | |
|---|---|---|---|---|---|---|
| | Mean µM | STD µM | CV% | Mean µM | STD µM | CV% |
| DISSOLVED OXYGEN | 196.07 | 0.13 | 0.07 | 198.84 | 0.14 | 0.07 |

Table 6: Results of duplicates. [1]AD=|duplicate #1 – duplicate #2|; [2] RPD%=Absolute
Difference *100/mean (dupl. #1, #2).

| Parameter | Range µM | mean Absolute Difference [1] µM | mean Relative Percentage Difference [2] |
|---|---|---|---|
| DISSOLVED OXYGEN | 179-240 | 0.18 | 0.09 |

**8 3.7 Nutrients (nitrite, nitrate, phosphate, and silicate), Total Dissolved Nitrogen
(TDN) and Total Dissolved Phosphorus (TDP).**
*Nutrients*
Analyses were performed at 40 °C on a four-channel, Quaatro SEAL Analytical Continuous
Flow Analyzer s/n 8014549; https://www.seal-analytical.com/Products/SegmentedFlow
Analyzers/QuAAtro39AutoAnalyzer/tabid/814/language/en-US/Default.aspx, according to
Hansen and Koroleff (1999). Nitrite was determined through the formation of a reddish-purple
azo dye, and measured at 520 nm (SEAL Method No. Q-030-04 Rev. 2). Nitrate was reduced
to nitrite in a copperized cadmium reduction coil and then determined as described for nitrite
(SEAL Method No. Q-035-04 Rev. 4). The determination of phosphate was based on the
reduced blue phospho-molybdenum complex, then measured at 880 nm (SEAL Method No. Q-
031-04 Rev. 1). Silicate was determined by means of acidic reduction of silicomolybdate to
molybdenum blue, then measured at 820 nm (SEAL Method No. Q-038-04 Rev. 0).



About 1400 nutrient samples were analyzed on board. The onboard precision of nutrient
measurements was determined on five replicates, at the beginning and at the end of the cruise.
The results are shown in table 7.
In addition, during the cruise 140 duplicates were analysed. The results are shown in table 8.
An internal quality check was daily performed by means of analyses of QUASIMEME samples
which provided results within the already certified ranges.
Table 7: On board precision of nutrient measurements

| Parameter | Beginning of the cruise | | | End of the cruise | | |
|---|---|---|---|---|---|---|
| | Mean μM | STD μM | CV% | Mean μM | STD μM | CV% |
| NITRITE (1) | 0.01 | 0.01 | 100 | 0.03 | 0.01 | 56.5 |
| NITRITE + NITRATE | 4.94 | 0.01 | 0.2 | 9.01 | 0.02 | 0.2 |
| PHOSPHATE | 0.18 | 0.01 | 5.5 | 0.41 | 0.01 | 3.1 |
| SILICATE | 8.34 | 0.03 | 0.3 | 9.55 | 0.04 | 0.5 |



Table 8: Analysis of duplicates. (1)AD=|duplicate #1 – duplicate #2|; (2) RPD%=Absolute
Difference *100/mean (dupl. #1, #2);   (3) Nitrite statistics was given just for completeness,
since the concentration levels recorded were too low, often below the detection limit.

| Parameter | Range<br><br>μM | mean Absolute<br>Difference (1)<br><br>μM | mean Relative<br>Percentage Difference<br>(2) |
|---|---|---|---|
| NITRITE (3) | 0-0.19 | 0.01 | 48.77 |
| NITRITE+NITRATE | 0.33-9.86 | 0.02 | 0.42 |
| PHOSPHATE | 0-0.47 | 0.01 | 5.13 |
| SILICATE | 0.93-11.00 | 0.04 | 0.72 |

***TDN and TDP***
About 550 samples for Total Dissolved Nitrogen and Total Dissolved Phosphorus (TDN and
TDP) on land-based laboratory analyses were collected and frozen at -20°C after filtration on
pre-combusted GF/F filter. The dissolved organic components, Dissolved Organic Nitrogen
(DON) and Dissolved Organic Phosphorus (DOP) were subsequently calculated by subtracting
their mineral constituents (NO3+NO2) and PO4, respectively.
**3.8 Discrete $CO_2$ System Measurements**
Discrete $CO_2$ variables were measured on board, being Dissolved Inorganic Carbon (DIC), pH,
Total Alkalinity (TA) and carbonate ion ($CO_3^{2-}$) at selected stations and depths (table 9). In
addition, discrete samples for DIC, pH and TA were analyzed specifically from surface Niskin
bottles to be compared with the continuous water supply feeding the underway partial pressure
of $CO_2$ (pCO2) system in determined stations. For further details see Hainbucher et al. (2018).



Table 9: Total number of $CO_2$ system samples analyzed during the MSM72 cruise. Total
number of fired bottles 1723.

|  | DIC | pH | TA | $CO_3^{2-}$ | Surface |
|---|---|---|---|---|---|
| **Samples** | 479 | 1160 | 949 | 391 | 22 |

### DIC

Samples for DIC were collected following transient tracers and dissolved oxygen, in 500 ml
borosilicate bottles following standard procedures. No poison was added. Samples were left at
room temperature in the dark until analysis, maximum 48 hours after collection. DIC samples
were analyzed with a MARIANDA VINDTA 3D system coupled with a UIC 5011 coulometer.
This analysis overall consists of extracting seawater $CO_2$ from a known volume of sample by
adding phosphoric acid, followed by coulometric detection (Johnson et al., 1993). No
calibration unit was available for the system. A new coulometric cell was prepared for every
batch of analysis and the accuracy of the DIC measurements was assessed by using Certified
Reference Material (CRM #158 & #170 provided by Prof. Dickson, UCSD). The calibration
factor obtained from the CRM was used for adjusting the final DIC of each sample measured
in the corresponding batch of analysis. In addition, substandard seawater (stabilized seawater
from the Cretan Sea 700m salinity minimum, stored in the dark in a 30 L container) was
analysed at the beginning and end of the batch analysis as a secondary quality control. The
precision of the DIC measurements was checked by: 1) double analysis from the same sample
and 2) replicate analysis from 4 to 5 samples collected from the same Niskin bottle. The
precision is estimated to be 1 $\mu$mol kg$^{-1}$ and the accuracy 2 $\mu$mol kg$^{-1}$.

### pH

Seawater spectrophotometric pH was measured following Clayton and Byrne (1993) at almost
all depths in the chemical and isotope stations during the MSM72 cruise (Table 1). This method
consists on adding a volume of indicator solution to the seawater sample, so that measuring the
absorbance of the sample at different wavelengths and obtaining the ratio between two of the
wavelengths absorbance is proportional to the sample pH. The indicator was a 2 mM solution
of unpurified m-cresol purple (Sigma Aldrich®) prepared in seawater and maintained at dark,
with no air contact (Absorbance Ratio 1.30). Samples were taken following standard procedures
immediately after DIC and directly into cylindrical 10 cm path length optical glass cells. The





cells were thermostatized at 25 ± 0.2ºC during one hour before analysis. Absorbance
measurements were obtained in the thermostated chamber of a double beam UV 2600 Shimadzu
spectrophotometer. The equipment was checked before the cruise for the absorbance and
wavelength accuracy using holmium standards. pH values on the total scale were calculated
and referred at 25°C by using the formula by Clayton and Byrne (1993). The injection of the
indicator in the sample slightly changes the sample pH. Following standard operating
procedures, double additions of the indicator were performed over a pH gradient in order to
obtain the corresponding correction (Hainbucher et al., 2018). The pH accuracy was controlled
measuring TRIS buffer solution samples (batch #72, provided by Prof. Dickson, UCSD). TRIS
samples were stabilized at three different temperatures covering the pH range found during the
MSM72 cruise. Differences between measured and theoretical TRIS pH varied between 0.009
to 0.005. The pH precision was checked by replicate analysis from cells collected at the same
Niskin from surface and deep waters. The precision is estimated to be 0.0004 pH units and the
accuracy 0.005 pH units. During the cruise some samples were also analyzed with purified m-
cresol purple provided by Prof. Byrne (USC).
***TA***
TA was analysed following a double end point potentiometric technique by Pérez and Fraga
(1987) further improved by Pérez et al. (2000). This technique is faster than the whole curve
titration, with comparable results (Mintrop et al., 2000). TA was measured at most stations and
depths (Table 1). Seawater samples for TA were collected after pH samples in 600 ml
borosilicate bottles following standard procedures. Samples were left at room temperature in
the dark until analysis, maximum 48 hours after collection. TA was measured by titration with
0.1 N hydrochloric acid dispensed with an automatic potentiometric titrator, Titrando
Metrohm®, provided with a combination glass electrode coupled with a temperature probe. The
electrode was standardized using a 4.41 pH ftatalate buffer made in $CO_2$ free seawater. The TA
accuracy was assessed with $CO_2$ CRM (batch #170, provided by Prof. Dickson, UCSD) In
addition to the CRM calibration, a drift control was conducted by analyzing substandard
seawater (big volume of seawater stored in the dark, as for DIC) at the beginning and at the end
of the analysis session. Each sample was measured twice and the mean value is reported, with
the mean standard deviation of all duplicate differences being 0.6 $\mu$mol kg$^{-1}$. In addition, typical
reproducibility analysis were performed from samples collected from the same niskin bottle at
different stations along the cruise. The TA precision is estimated to be 1 $\mu$mol kg$^{-1}$ and the
accuracy 2 $\mu$mol kg$^{-1}$.



*$CO_3^{2-}$*
The $CO_3^{2-}$ ion concentration was determined spectrophotometrically following Byrne and Yao
(2008) incorporating the recent improvements by Patsavas et al. (2015), at selected stations and
depths (Table 1) Samples for $CO_3^{2-}$ were collected after TA following the same procedure as
for pH but within cylindrical optical quartz 10 cm pathlength cuvettes. The cells were stabilized
at 25°C for one hour before the analysis, maximum 24 hours after collection. A solution of
0.022 M of $Pb(ClO_4)_2$ was added to the seawater sample and the $PbCO_3$ complex formed
afterwards was detected spectrophotometrically in the UV spectra. Absorbance measurements
were obtained in the thermostated chamber of a double beam UV 2600 Shimadzu
spectrophotometer. The equipment was checked before the cruise for the absorbance and
wavelength accuracy width using holmium standards. The $CO_3^{2-}$ in µmol kg$^{-1}$ is the
concentration of ion carbonate at 25ºC calculated using the formula by Patsavas et al.
(2015). The $CO_3^{2-}$ precision was checked by replicate analysis from cells collected at the same
niskin from surface and deep waters. It is estimated to be 1 µmol kg$^{-1}$.
## 3.9   Measurements of CFC-12 and SF$_6$
During the cruise, one gas chromatograph purge-and-trap (GC/PT) system was used for the
measurements of the transient tracers CFC-12 and SF$_6$. The system is modified versions of the
set-up normally used for the analysis of CFCs (Bullister and Weiss, 1988). All samples were
collected in 250 mL ground glass syringes, of which an aliquot about 200 mL was injected to
the purge-and-trap system, normally within 5 hours from sampling.
The traps consisted of 100 cm 1/16” tubing packed with 70cm Heysep D kept at temperatures
between -70 and -75°C during trapping. The traps were desorbed by heating to 120°C and
passed onto the pre-column.  The pre-column consisted of 20 cm Porasil C followed by 20 cm
Molsieve 5A in a 1/8” stainless steel column. The main column was a 1/8” packed column
consisting of 180 cm Carbograph 1AC (60-80 mesh) and a 50 cm Molsieve 5A post-column.
Both columns were kept isothermal at 60°C. Detection was performed on an Electron Capture
Detector (ECD).
Standardization was performed by injecting small volumes of gaseous standard containing
CFC-12 and SF$_6$. This working standard was prepared by the company Dueste-Steiniger (DS1,).
The CFC-12 and SF$_6$ concentrations in the working-standard has been calibrated vs. a reference
standard obtained from R.F Weiss group at SIO, and the CFC-12 data are reported on the SIO98
scale. Calibration curves were measured roughly once a week in order to characterize the non-
linearity of the system, depending on workload and system performance. Point calibrations
were always performed between stations to determine the short-term drift in the detector.
Replicate measurements were taken except for near coastal stations due to high workload. To
assess the reproducibility of the set-up, 50 replicates samples were run, and resulted in a
reproducibility of 1.0 % or 0.01 pmol kg$^{-1}$ for CFC-12 and 2.3% or 0.03 fmol kg$^{-1}$ for SF$_6$. In
total we successfully measured 1084 samples on 68 stations for transient tracers.
In addition to the on-board analysis, on three stations (#52, #84, and #106) 1500 ml glass
ampoules were flame sealed for later analysis in the lab in Kiel for the detection of novel
halogenated tracers such as HFC134a and HCFC22 (Li and Tanhua, 2019).
**3.10 Dissolved Organic Carbon (DOC)**
Seawater samples for DOC were collected from the CTD-Rosette into 250 ml Polycarbonate
Nalgene bottles. Samples were filtered through a 0.2 μm Nylon filter under high-purity air
pressure. Filtered samples were collected in 60 ml Nalgene bottles, acidified and stored at 4°C
and in the dark.
DOC measurements were carried out with a Shimadzu Total Organic Carbon analyzer (TOC-
Vcsn), by high temperature catalytic oxidation. Samples were acidified with HCl 2N and
sparged for 3 minutes with $CO_2$-free pure air, in order to remove inorganic carbon. From 3 to 5
replicate injections were performed until the analytical precision was lower than 1% (± 1μM).
A five-point calibration curve was done by injecting standard solutions of potassium hydrogen
phthalate in the expected concentration range of the samples. At the beginning and end of each
analytical day the system blank was measured using low carbon water (LCW) and the reliability
of measurements was controlled by comparison of data with a DOC reference (CRM) seawater
sample kindly provided by Prof. D.A. Hansell of the University of Miami
(http://yyy.rsmas.miami.edu/groups/biogeochem/CRM.html).
In total 650 samples were collected in 38 stations. Samples were collected at the following
depths: 10, 25, 50, 75, 100, 150, 200, 300, 400, 500, 750, 1000 and every 250 m until the
bottom.
**3.11 Chromophoric dissolved organic matter (CDOM)**
Chromophoric dissolved organic matter (CDOM) is the fraction of DOM that absorbs light at
visible and ultraviolet (UV) wavelengths. It plays a key role in the marine ecosystem by





regulating light penetration into the water column (Nelson and Siegel, 2013) and preventing cellular DNA damage (Herndl et al., 1993; Häder and Sinha, 2005). A fraction of CDOM re-emit part of the absorbed light and is called fluorescent DOM (FDOM). The study of the absorption properties of CDOM, together with the analysis of the excitation-emission matrixes (EEMs) through the parallel factorial analysis (PARAFAC) can give qualitative information on the different groups of chromophores (protein-like, humic-like and PAH-like) present in the DOM pool, their changes due to photodegradation and/or microbial transformation, the main sources of CDOM and an indirect estimation of its molecular weight and aromaticity degree (Stedmon and Nelson, 2015, Retelletti et al., 2015, Gonelli et al., 2016, Margolin et al., 2018). The CDOM data collected during the MSM72 cruise will represent an unique opportunity to: (i) Compare CDOM optical properties in the different water masses of the Mediterranean Sea with those collected in the Geotraces cruise (Spring-summer 2013) and to relate them to the different trophic conditions of the basin; (ii) Study the relationship between DOC and CDOM in the surface, intermediate and deep waters.

### 3.12 Sampling for Measurements of Stable Carbon Isotopes on Dissolved Inorganic Carbon (DIC)

Samples for the determination of stable carbon isotopes ($\delta^{13}C$) of Dissolved Inorganic Carbon (DIC) were taken on 11 stations (the "isotope stations", normally performed as a double cast) in the various basins along the cruise-track. In total 214 samples were taken in 100 ml dark glass bottles immediately poisoned with 100 µL saturated mercury chloride. The samples were measured off-line during fall of 2018 at the Centre for Isotope Research (CIO), Energy and Sustainability Research Institute Groningen (ESRIG), University of Groningen.

### 3.13 NO$_3^-$ isotopes ($\delta^{15}N$ & $\delta^{18}O$)

Samples for nitrogen (N) and oxygen (O) isotopes in nitrate (NO$_3^-$) and nitrate+nitrite (NO$_3^-$+NO$_2^-$) analysis were collected at 44 stations evenly distributed along the transect. In total, 790 samples have been collected. High-resolution NO$_3^-$ $\delta^{15}N$ and $\delta^{18}O$ measurements represent a powerful tool to unravel the sources and sinks of reactive (i.e., fixed) N at the scale of the Mediterranean Sea. Complemented with coral-bound $\delta^{15}N$ records covering the last centuries, these measurements may also shed light on the contribution of industrially fixed N to the reactive N budget, by revealing the large-scale systematics required to interpret the records back in time.



Unfiltered samples for N and O isotopic composition of $NO_3^-$ were collected in 60 mL plastic bottles and stored frozen (-20°C) until analysis. $NO_3^-+NO_2^-$ $\delta^{15}N$ and $\delta^{18}O$ will be measured (2019-2020) at the Max Planck Institute using the denitrifier method (Sigman et al., 2001; Casciotti et al., 2002). Briefly, 3-20 nmol of $NO_3^-+NO_2^-$ is quantitatively converted to $N_2O$ gas by denitrifying bacteria (*Pseudomonas aureofaciens*) that lack an active $N_2O$ reductase. The $N_2O$ is then analysed by gas chromatography-isotope ratio mass spectrometer (GC-IRMS; MAT253, Thermo) with on-line cryo-trapping (Weigand et al., 2016). Measurements are referenced to air $N_2$ for $\delta^{15}N$ and VSMOW for $\delta^{18}O$ using the nitrate reference materials IAEA-NO3 and USGS-34. For $NO_3^-$ $\delta^{15}N$ and $\delta^{18}O$ analysis, $NO_2^-$ is removed with the sulfamic acid method prior to the isotopic analysis (Granger and Sigman, 2009). The reproducibility is generally better than 0.1‰ for $\delta^{15}N$ and $\delta^{18}O$, respectively.

### 3.14 LISST – DEEP

The LISST-Deep instrument obtains in-situ measurements of particle size distribution, optical transmission, and the optical volume scattering function (VSF) at depths down to 3,000 meters. It is manufactured by Sequoia Inc., and owned by the Hellenic Centre for Marine Research (HCMR) – Greece.

Using a red 670nm diode laser and a custom silicon detector, small-angle scattering from suspended particles is sensed at 32 specific log-spaced angle ranges. This primary measurement is post-processed to obtain sediment size distribution, volume concentration, optical transmission, and volume scattering function. The LISST-Deep s/n 4004 is categorized as a type B instrument, which means that the range of particles it measures ranges from 1.25 μm to 250 μm. The LISST-Deep must be powered externally at all times. This is typically achieved by connecting it to a rosette, getting power from the main CTD unit.

Parameters measured during the cruise were:

- Particle size distribution from 1.25-250μm or 2.5-500μm
- Depth (3000 m max depth @ 0.8 m resolution)
- Optical transmission @ 0.1 % resolution
- Beam attenuation Coefficient @ 0.1 $m^{-1}$ resolution
- Volume concentration @ 0.1 μl/l resolution
- Volume scattering function (VSF)



The measurement of these parameters provided important information on the number, size and
quality (phytoplankton, sediment, etc.) of the suspended matter in the water column. Further
information for the determination of water masses was provided by the estimation of the
intrinsic optical properties. Finally, for the first ~ 100m we estimated the color of the sea and
compared this estimation with satellite images, providing valuable information for the
calibration of satellite algorithms.
For the cruise MSM72 the sampling of these optical estimates is in itself an important
achievement because, for the first time LISST – DEEP was used to record data in a transect
over the full length of the Mediterranean Sea. Furthermore, the estimation of these parameters
combined with POC - PON estimation, and other physical and chemical parameters, improve
the study of the dynamics of the Mediterranean Sea.
In general, the use of LISST – DEEP during the cruise follows the standard methods which are
provided by Sequoia Inc, but with one important difference. For the estimation of the above
parameters the use of a background file is required for normalization purposes. This file is
normally produced in laboratory conditions with mili – q 2 filtered water. However, experience
until now has proved that especially in the eastern Mediterranean Sea (which is characterized
as ultra-oligotrophic) the use of this background file leads us to an overestimation of the
parameters and especially of the beam attenuation coefficient. Therefore, during this cruise we
used a sampled in situ background file chosen as the minimum of the sum of the digital counts
in the 32 rings and where the LaserPower to LaserReference (Lp/Lr) ratio is maximum.
The main problem which we faced was the frequent change of the CTD main unit and the
different cables that we had to use for the instrument connection to the CTD. Fortunately, with
the most valuable help of the cruise technician we managed to deploy the LISST – DEEP as
much as possible. Additionally, the maximum depth limitation of the instrument (3000m)
enforced us to remove it in deep casts achieving a total of 54 stations.
**4.      Discussion and Conclusion**
Discussion and conclusion will focus in this publication on the quality of the data of MSM72
cruise. We will concentrate here on the basic physical and biogeochemical parameters, as
selected examples, to show the relevance of the sampled data and so as to be able to answer the
questions on the scale and variability of the circulation and biogeochemical cycle in the
Mediterranean Sea (see Introduction).

## 4.1 Physical parameters

The west east section (figure 2) is a typical example for the distribution of temperature and
salinity in the Mediterranean Sea showing the different heat and salt content between the
western and eastern basin. A clear intrusion of the salty Levantine Intermediate Water (LIW)
from east to west in the first 500m is depicted while the low salinity Atlantic Water (AW)
protrudes eastwards creating a front at about 20-22°E.

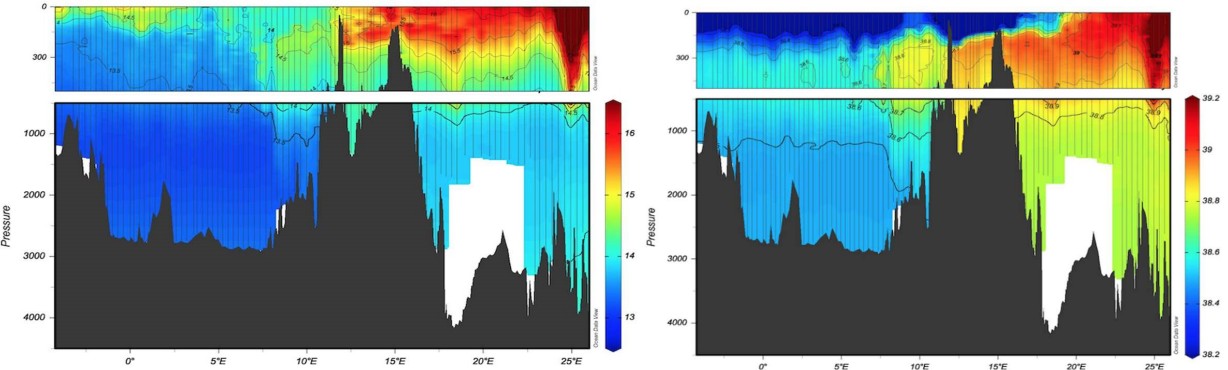

Figure 2: West-east temperature (left) and salinity (right) sections through the Mediterranean
Sea.
The underway CTD data are a valuable addition to the classical CTD data. They enhance the
resolution of data in the horizontal scale and give insight in eddy activity. Although the data do
not reach to the bottom, the vertical resolution with about 1000 m is useful to characterize scales
relevant for the LIW transport.
The uCTD salinity distribution of figure 3, located along the easternmost part of the northward
transit in the Ionian Sea, shows that the Pelops gyre is well resolved.

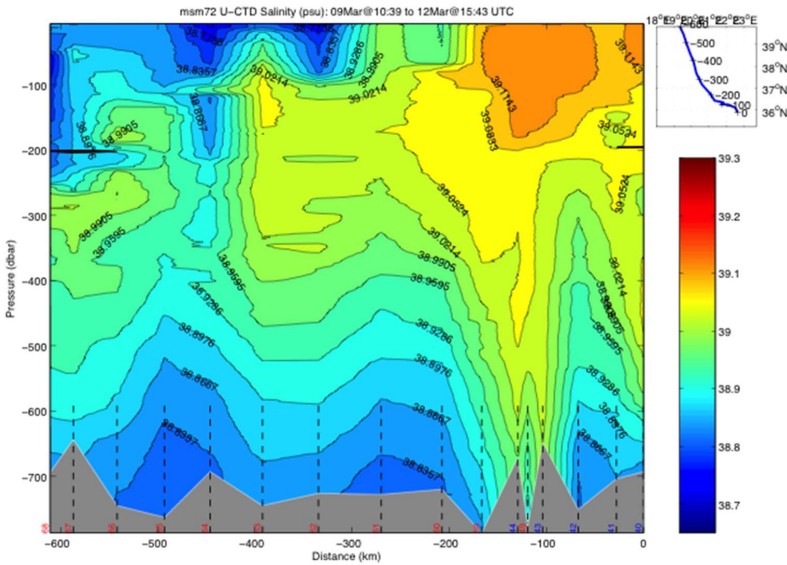

Figure 3:  uCTD salinity transect. Location is shown in the upper right panel.
Considering the route of the ship during the cruise, it was possible to identify different ADCP
transects that correspond to areas with the most important water mass dynamics. In particular
the most important sections were: gyre activity in the area west of Crete and south of
Peloponnese, the west Cretan, Otranto (figure 4) and Sicily Straits, the east boundary of the
Ionian Sea and the west-east Mediterranean transect. The north-south current component (figure
4) in the Strait of Otranto clearly shows the outflow of the Adriatic Deep Water (AdDW) along
the western part while in the upper and intermediate layer of the central part the inflow of the
Levantine Intermediate Water (LIW) proceeds.

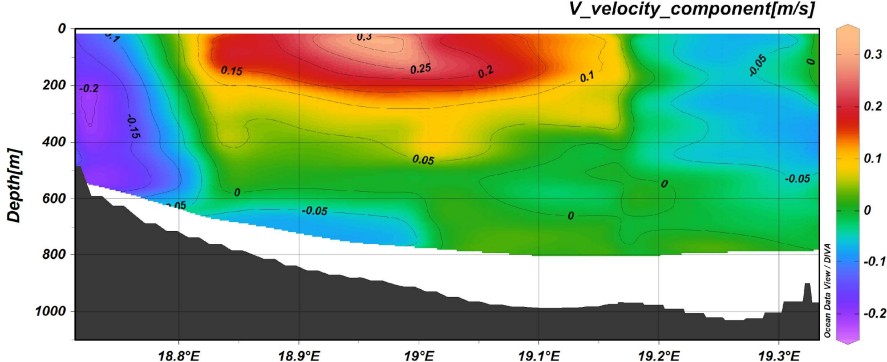

Figure 4: Transect across the Strait of Otranto from ADCP 38, positive numbers correspond to
northward currents.



### 4.2 Biogeochemical parameters

The vertical distribution of dissolved oxygen along a section from the Cretan Sea to Gibraltar,
including part of the Cretan Passage and the southern Ionian is shown in figure 5. This section
shows the Oxygen Minimum Layer (<180 μmoles/kg) which occupies the layer 500-1500m.
Increased oxygen towards the bottom indicate the ventilation of deep water in the
Mediterranean.   The western part of the Ionian Sea appears to be better oxygenated than the
eastern part due to the spreading of newly ventilated dense water from the Adriatic Sea via the
Otranto Strait Strait – a feature that is observed in the transient tracer section as well.

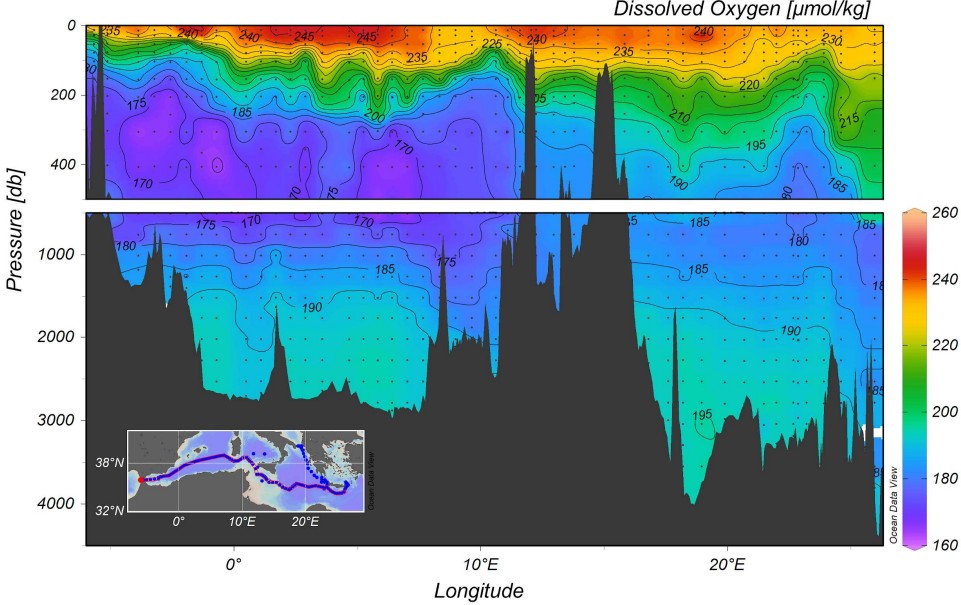

Figure 5: Distribution of dissolved oxygen along the trans-Mediterranean section.
Figure 6 illustrates the distribution of nitrate along the quasi-zonal section. Interesting features
include: the maximum nutrient layer in the range of depth of 500-1500 m which is co-located
to the minimum of transient tracers; the deepest layer shows an homogeneous distribution of
nutrients and the nutrient impoverished upper layer is, not yet completely depleted of nutrients,
likely do to subject to mesoscale dynamics (as, for example, south of Crete).

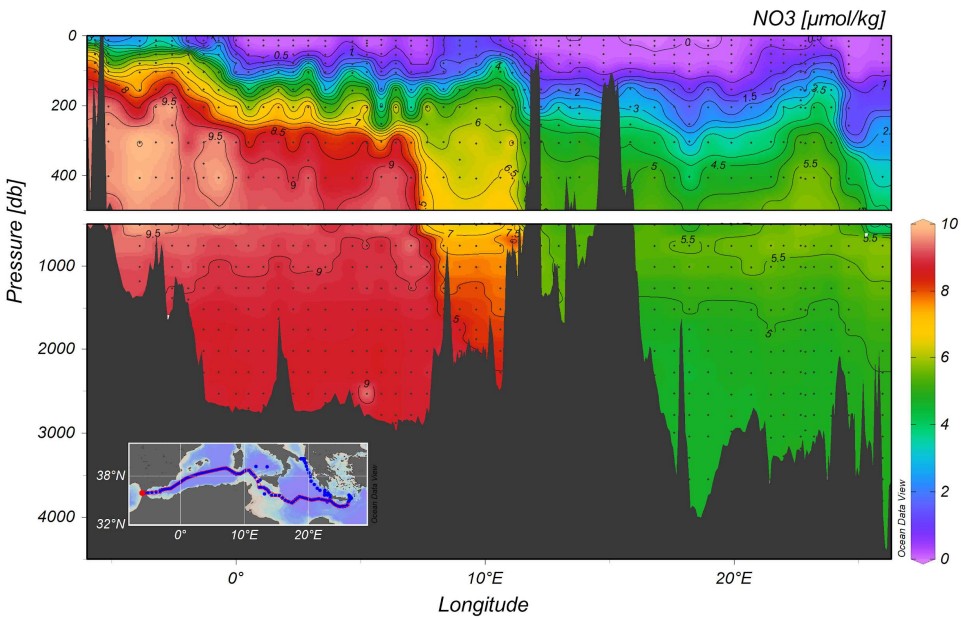

Figure 6: Distribution of nitrate along the trans-Mediterranean section.

The DOC data collected during the MSM72 cruise represents an unique opportunity to (i)
investigate the long-term variation in DOC distribution in intermediate and deep waters on a
basin scale; (ii) quantify the role of DOC in C export and sequestration in the Mediterranean
Sea; (iv) estimate DOC mineralization rates; (v) asses the functioning of microbial loop in the
different areas of the Mediterranean Sea.
DOC concentrations range between 34 and 80 µM (figure 7). The highest values (> 50 µM )
were observed in the upper 200 m, with a marked increase moving eastward. This feature has
already been observed in the Med Sea (Santinelli, 2015; Santinelli et al., 2012) and can be
explained by different processes such as nutrient limitation of heterotrophic prokaryotes
growth, the occurrence of recalcitrant DOC that is not available for consumption. The high
stratification, occurring in the easternmost stations, makes DOC accumulation more visible. A
different functioning of the microbial loop has been reported for the western and eastern
Mediterranean Sea and these data support that DOC dynamics in the surface layer of the two
sub-basins is different . The lowest concentrations (< 40 µM) are between 1000 and 2000 m, in
the bottom waters a slight increase in DOC can be observed. This feature, already reported for
the Mediterranean Sea, can be explained by the export of the DOC accumulated in the surface
layer by deep water formation (Santinelli, 2015 and references herein).



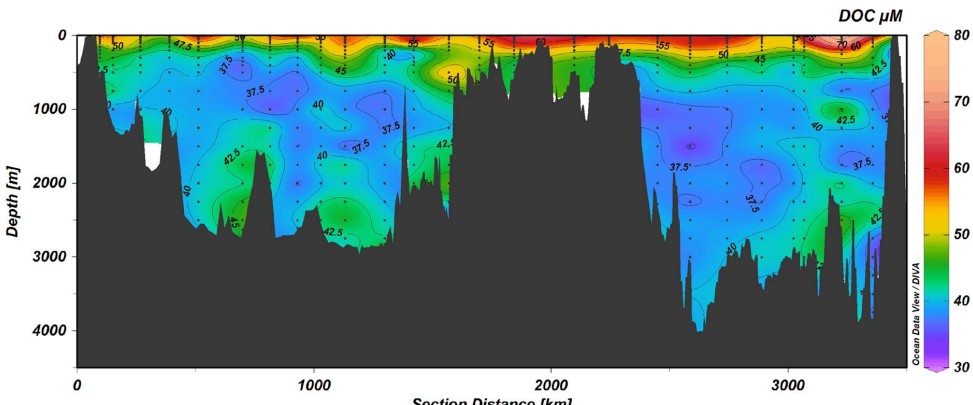

Figure 7: DOC vertical distribution along the trans-Mediterranean section

## 5. Data access

Data are published at the information system PANGAEA and CCHDO;

## 6. Acknowledgements

We thank Captain Björn Maaß, his officers and the crew of R/V MARIA S. MERIAN for the
support of our scientific programme, for their unending competent and friendly help.
The financial support for the cruise was provided by the project of the "Deutsche
Forschungsgemeinschaft" U4600DFG040204. We gratefully acknowledge their support.
The $CO_2$ team was funded by an internal IEO grant MEDSHIP18, A.E.R. Hassoun was funded
by a POGO grant. The OGS team was funded by an internal grant.

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
