# Peer review of "Physical and Biogeochemical Parameters of the"

_Earth System Science Data, 2020_

## Referee Comment (RC1) · Anonymous Referee #1 · 10 Aug 2020

The MedShip program (GO-SHIP sections in the Mediterranean Sea) supports the investigation of its relevant objectives which are, amongst others, engaged in the determination of changes and of long-term variability of hydrographic and biogeochemical parameters in the Mediterranean Sea. This survey is a valuable contribution to improve the database of the Mediterranean Sea for a better understanding of the variability on multiple timescales and for numerical model evaluations.

Abstract:

[Figure]

- Page 1 line 30: "internal processes". Detail the meaning of "internal" (convection/mixing, ventilation others ?)

- Page 2 line 3: "we report on data". Remove the "on"

- Say somewhere in the introduction that this cruise is part of the MedShip long-term repeat cruise section that is conducted every 5 years in Mediterreanean Sea to observe changes and impacts on physical and biogeochemical variables.

- Table 1a: add the name of DO sensors: SBE43 others ? $\mu$mol/l or $\mu$mol/kg ? (table 1a vs table 1b). Remove the depth (m) variable which is deduced from the pressure sensor

Introduction:

- Page 5 lines 1 to 3: the sentence is not entirely true. The interannual variability of convection events is significant in the western Mediterranean Sea due to the variability of atmospheric forcing (wind intensity and duration) and the preconditioning event (LIW characteristics). In this context, some years are more convective than others, which has a different impact on the characteristics of the water masses.

Data provenance:

- Figure 1: the map is too small and this is impossible to distinguish the different plots

Methods:

- Table2: add the calibration date of the sensors

- Paragraph 3.1: indicate at which depths the salinity samples have been collected

- Section 3.6: authors do not mention the oxygen from the SBE43 sensor which is used now for 20 years. With Winkler measurements, the SBE43 can be corrected through the least square method which is used to adjust the calibration coefficients precisely (+/- 2 $\mu$mol/kg). In this case, it is recommended to sample once day the

entire water column for Winkler analysis. This procedure does not affect the accuracy of O2 measurements. The authors should now have confidence in the SBE43 sensor as they do with the optode used for surface measurements. The strategy of collecting Winkler samples at all stations therefore seems to me inappropriate now. Indicate clearly at which depths the samples have been collected for Winkler analysis.

- Table 9 has to be rearranged (the text is illegible)

Discussions:

- Figure 2: indicate the name of straits etc.. . . on the figure. Do as the figure 6 with the map of the cruise which will help to visualize the location of the T and S sections

- Figure 3: indicate the locations. It's difficult to understand where uCTD has been performed. Distance from where ?

- Page 26 line 10: replace "Med Sea" with "Mediterranean Sea"

- Figure 7: add the section map

―――――――――――――――――――

---

## Referee Comment (RC2) · Anonymous Referee #2 · 17 Aug 2020

Journal: Earth System Science Data

Title: Variability and Trends in Physical and Biogeochemical Parameters of the Mediterranean Sea during a Cruise with RV MARIA S. MERIAN in March 2018

Manuscript ID: essd-2020-82

In this manuscript, a survey on long-term variability and trends in physical and biogeochemical parameters of the Mediterranean Sea during a cruise conducted in March 2018 with RV MARIA S. MERIAN, was carried out. The investigation was conducted

with the aim to assess the hydrographycal situation after the major climatological shifts in the eastern and western part of the Mediterranean basin.

Abstract

Page 2 line 1: ". . . variability can be influenced by . . ." change to "variability of this system can be influenced by . . .";

Methods

Page 7 lines 4-5: Figure 1, "yellow squares" are difficult to detect in the figure 1;

Page 15 line 11: "NO3, NO2, PO4" change to subscript;

Page 16 Table 9: correct letters and numbers disposition;

Page 16 line 28: "m-cresol" change m- to Italic style;

Page 21 line 9: "NO3" change to 3 subscript "NO3".

Discussion and Conclusion

Page 23 Figure 2: numbers inside figure 2 are difficult to read;

Page 25 line 10: "Otranto Strait Strait" change to "Otranto Strait";

Page 26 lines 8-19: ". . . nutrient limitation of heterotrophic prokaryotes growth, . . .". Previous investigations indicate that the Mediterranean Sea is a P-limited system, especially in the eastern basin, and that phosphate availability may limit both primary production and heterotrophic bacteria. Analyses of heterotrophic microbial communities conducted during cruise investigations, have been included in some investigations, giving insights on DOC consumption by heterotropic prokaryotes, and report DOC data combined with microbiological information. The metabolism of the dominant heterotrophic and photoheterotrophic bacterial plankton populations is synchronized to autotrophic processes. Little is known about how microorganisms degrade and metabolize this pool of organic nutrients. Isolation and characterization of bacterial strains

obtained from native water samples collected during cruise, could add information on the role of different strains of bacterial community in degrading DOC.

---

## Author Response (AR1)

**1 Response to reviewer #1**

- 2 We thank reviewer #1 for all his/her comments and criticism which helped us to improve our 3 manuscript.
- 4 All remarks were taken into account and the text and figures were amended accordingly. There is one
- 5 exception: We did not remove the depth (m) variable in table 1a). This variable is not a parameter
- 6 given by the CTD but belongs to the output of the ADCP and IADCP, where no pressure values are7 given.
- 8 In general, we agree with the arguments of the reviewer on oxygen measurements. In our particular
- 9 case, we had a lot of technical problems with the CTD system which led to a loss of data quality of the
- 10 CTD oxygen values. The high-resolved measured oxygen data were an equivalent alternative. We now
- 11 comment on this in the manuscript.
- 12

**13 **Response to reviewer #2**

14 We thank reviewer #2 for all his/her comments and criticism which helped us to improve our 15 manuscript.

- 16 All remarks were taken into account and the text and figures were amended accordingly.
- 17 For the DOC part, the reviewer gives some very helpful insight to the biology and chemistry of the
- 18 Eastern Mediterranean. This is certainly worth taking into consideration for a scientifically oriented
- 19 paper analyzing the data. This is a paper for ESSD and though, the focus of the paper is the presentation
- 20 of the data itself and a discussion of its quality. We keep the reviewer's comments in mind for further
- 21 publications on the scientific results.
- 22

**Variability and Trends in Physical and Biogeochemical Parameters of the Mediterranean Sea during a Cruise with RV MARIA S. MERIAN in March 2018**

4

Dagmar Hainbucher1, Marta Álvarez4, Blanca Astray Uceda4, Giancarlo Bachi5,
Vanessa Cardin3, Paolo Celentano6, Spyros Chaikakis7, Maria del Mar Chaves
Montero3,9, Giuseppe Civitarese3, Noelia M. Fajar 4, Francois Fripiat10, Lennart
Gerke2, Alexandra Gogou7, Elisa Fernández Guallart4, Birte Gülk1, Abed El
Rahman Hassoun8, Nico Lange2, Andrea Rochner1, Chiara Santinelli5, Tobias
Steinhoff2, Toste Tanhua2, Lidia Urbini3, Dimitrios Velaoras7, Fabian Wolf2,
Andreas Welsch1

- 12 [1] {Institut für Meereskunde, CEN, Universität Hamburg, Bundesstraße 53, 20146 Hamburg,
- 13 Germany}
- 14 [2]{GEOMAR, Helmholtz-Zentrum für Ozeanforschung Kiel, Wischhofstr. 1-3, 24148 Kiel,
- 15 Germany}
- 16 [3]{Istituto Nazionale di Oceanografia e di Geofisica Sperimentale OGS, Dept. Of
- 17 Oceanography, Borgo Grotta Gigante 42/c, 34010 Sgonico, Trieste, Italy}
- 18 [4] {Instituto Español de Oceanografía (IEO), centro de A Coruña, Spain}
- 19 [5]{Istituto di Biofisica, CNR, Pisa, Italy}
- 20 [6] {Istituto di Scienze Marine, Venezia, Italy}
- 21 [7] {Helenic Centre for Marine Research, Athens, Greek}
- 22 [8] {National Council for Scientific Research in Lebanon. National Center for Marine Sciences}
- 23 [9]{Centro Euro-Mediterraneo sui Cambiamenti Climatici CMCC, Bologna, Italy}
- 24 [10]{Max Planck Institute for Chemistry, Mainz, Germany}
- 25 *Correspondence to:* Dagmar Hainbucher (dagmar.hainbucher@uni-hamburg.de)
- 26
- 27

**1 Abstract**

2 The last decades have seen dramatic changes in the hydrography and biogeochemistry of the 3 Mediterranean Sea. The complex bathymetry, highly variable spatial and temporal scales of atmospheric forcing, convective and ventilation processes contribute to generate complex and 4 unsteady circulation patterns and significant variability in biogeochemical systems. Part of the 5 variability of this system can be influenced by anthropogenic contributions. Consequently, it is 6 7 necessary to document details and to understand trends in place to better relate the observed processes and to possibly predict the consequences of these changes. In this context we report 8 9 data from an oceanographic cruise in the Mediterranean Sea on the German research vessel MARIA S. MERIAN (MSM72) in March 2018. The main objective of the cruise was to 10 contribute to the understanding of long-term changes and trends in physical and biogeochemical 11 parameters, such as the anthropogenic carbon uptake and to further assess the hydrographical 12 situation after the major climatological shifts in the eastern and western part of the basin, known 13 as the Eastern and Western Mediterranean Transients. During the cruise, multidisciplinary 14 measurements were conducted on a predominantly zonal section throughout the Mediterranean 15 Sea, contributing to the global GO-SHIP repeat hydrography program, and particularly to its 16 17 Mediterranean Sea component, Med-SHIP, and adhering to the GO-SHIP requirements. 18

**19 Data coverage and parameter measured**

20 Repository-Reference (table 1a and table 1b):

21

1 Table 1a. List of physical parameters from MARIA S. MERIAN cruise MSM72 as seen in the

| Parameter Name          | Short name | Unit                 | Method                                  | Comment |
|-------------------------|------------|----------------------|-----------------------------------------|---------|
| DATE/TIME               | Date/Time  |                      |                                         | Geocode |
| LATITUDE                | Latitude   |                      |                                         | Geocode |
| LONGITUDE               | Longitude  |                      |                                         | Geocode |
| Pressure, water         | Press      | dbar                 | CTD, SEA_BIRD SBE 911plus               |         |
| Temperature, water      | Temp       | °C                   | CTD, SEA_BIRD SBE 911plus               |         |
| Salinity                | Sal        |                      | CTD, SEA_BIRD SBE 911plus               | PSU     |
| Oxygen                  | 02         | <mark>µmol/kg</mark> | CTD with attached oxygen                |         |
|                         |            |                      | sensor <mark>(SBE43)</mark> calibrated, |         |
|                         |            |                      | corrected using Winkler                 |         |
|                         |            |                      | titration                               |         |
| Pressure, water         | Press      | dbar                 | Underway CTD (UCTD),                    |         |
|                         |            |                      | Oceanscience                            |         |
| Temperature, water      | Temp       | °C                   | Underway CTD (UCTD),                    |         |
|                         |            |                      | Oceanscience                            |         |
| Salinity                | Sal        |                      | Underway CTD (UCTD),                    | PSU     |
|                         |            |                      | Oceanscience                            |         |
| DEPTH, water            | Depth      | m                    |                                         |         |
| Current velocity        | UC         | m/s                  | Shipboard Acoustic Doppler              |         |
| east-west               |            |                      | Current Profiling (SADCP)               |         |
| Current velocity | VC         | m/s                  | Shipboard Acoustic Doppler              |         |
| north-south             |            |                      | Current Profiling (SADCP)               |         |
| DEPTH, water            | Depth      | m                    |                                         |         |
| Current velocity        | UC         | m/s                  | lowered Acoustic Doppler                |         |
| east-west               |            |                      | Current Profiling (lADCP)               |         |
| Current velocity | VC         | m/s                  | lowered Acoustic Doppler                |         |
| north-south             |            |                      | Current Profiling (IADCP)               |         |

2 PANGAEA database. PI: Dagmar Hainbucher

3

- 1 Table 1b. List of biogeochemical parameters from MARIA S. MERIAN cruise MSM72 as
- 2 seen in the CCHDO database. PI: Toste Tanhua

| Variable                                                    | Unit                  |
|-------------------------------------------------------------|-----------------------|
| Dissolved Oxygen (O 2 )                          | µmol kg-1             |
| Sulphurhexafluorid (SF 6 )                       | fmol kg -1 |
| CCl 2 F 2 (CFC-12)                    | pmol kg-1             |
| Nitrate (NO 3 -)                                 | µmol kg-1             |
| Nitrite (NO 2 -)                                 | µmol kg-1             |
| Phosphate (PO4 2- )                              | µmol kg-1             |
| Silicate (Si)                                               | µmol kg-1             |
| Dissolved Inorganic Carbon (DIC)                            | µmol kg-1             |
| Total Alkalinity (TA)                                       | µmol kg-1             |
| рН                                                          | Total scale @ 25°C    |
| Carbonate (CO 3 2- )                  | µmol kg-1             |
| $\delta^{13}$ C of DIC                                      | Per mille             |
| Total Dissolved Nitrogen (TDN)                              | µmol kg-1             |
| Total Dissolve Phosphorus (TDP)                             | µmol kg-1             |
| CHClF 2 (HCFC-22)                                | pmol kg -1 |
| C 2 H 3 Cl 2 F (HCFC-141b) | pmol kg -1 |
| C 2 H 3 ClF 2 (HCFC-142b)  | pmol kg -1 |
| CH 2 FCF 3 (HFC-134a)                 | pmol kg -1 |
| C 2 HF 5 (HFC-125)                    | pmol kg-1             |
| CHF3 (HFC-23)                                               | pmol kg -1 |

- 1https://doi.pangaea.de/10.1594/PANGAEA.905902(for CTD)2https://doi.pangaea.de/10.1594/PANGAEA.913512(for UCTD)3https://doi.pangaea.de/10.1594/PANGAEA.913608(for ADCP)4https://doi.pangaea.de/10.1594/PANGAEA.913505(for IADCP)5https://doi.pangaea.de/10.1594/PANGAEA.905887(for chemical data)
- 6 https://doi.org/10.25921/z7en-hn85
- 7 A link to the summary page of the cruise MSM72 can be found in the PANGAEA data base

(for pCO2)

- 8 under: https://www.pangaea.de/?q=msm72&f.campaign%5B%5D=MSM72
- 9 Coverage:  $34^{\circ}$ N- $41^{\circ}$ N,  $6^{\circ}$ W- $28^{\circ}$ E
- 10 Location Name: The Mediterranean Sea
- 11 Date/Time Start: 2. March 2018
- 12 Date/Time End: 3. April 2018
- 13

**14 **1.** Introduction**

15 Contrary to earlier ideas that the Mediterranean Sea is always in a steady state, we now know in the light of new research that the Mediterranean Sea is not and it is potentially sensitive to 16 17 climatic changes (Malanotte-Rizzoli, 2014). Proof of this are the drastic changes that the eastern Mediterranean (EMed) has undergone in the past. The largest climatic event, named Eastern 18 Mediterranean Transient (EMT), occurred in the EMed between the late 1980's and early 19 1990's, where deep-water formation switched from the Adriatic to the Aegean Sea. This 20 21 episode modified the thermohaline characteristics of the outflow through the Sicily Channel, advecting anomalously salty and warm Levantine Intermediate Water (LIW) to the western 22 Mediterranean Sea (WMed) and leading to a significant increase in temperature and salt in the 23 intermediate and deep layers of the WMed. Additionally, strong deep convection induced by 24 extreme atmospheric events during winter time 2004-2006 (low precipitation, cold, persistent 25 winds) was also enhancing salt and temperature in the entire basin up to about 1600 m 26 (Schroeder et al., 2006, Schroeder et al., 2008). This abrupt climate shift is referred to as 27 Western Mediterranean Transient (WMT) and the physical changes are comparable to the EMT, 28 both in terms of intensity and observed effects (Schroeder et al., 2008). The existence of both 29 transients contradicts the hypothesis of a steady state. On the other hand, it has also been proven 30 31 that an EMT has never been observed before (Roether et al., 2013).

The characteristic of the Mediterranean Sea is also such that it has the potential to sequester large amounts of anthropogenic CO2, Cant, since the Mediterranean Sea has high alkalinity and temperature, which can be rapidly transported to deep by the overturning circulation (e.g. Schneider et al., 2010). The column inventories of Cant in the Mediterranean are among the highest found in the world oceans; the Mediterranean Sea thus stores a significant portion of the global anthropogenic emissions of Cant despite its relatively small volume.

7 Furthermore, marine dissolved organic carbon (DOC) represents the largest reservoir of reduced carbon ( $662 \cdot 10^{15}$  g C) on Earth (Hansell, 2009), it therefore plays a major role in the 8 global carbon cycle. Its role in the functioning of marine ecosystems is equally crucial since 9 10 DOC is released at all the levels of the food web, as a byproduct of many trophic interactions and/or metabolic processes and is the main source of energy for the heterotrophic prokaryotes 11 12 (Carlson and Hansell, 2015). Although most of DOC is produced in-situ, external sources (atmosphere, rivers, sediments) may affect its concentration and distribution. Physical 13 processes, such as deep-water formation, thermohaline circulation, vertical stratification and 14 mesoscale activities have been reported to be the main drivers of DOC distribution in the 15 Mediterranean Sea (Santinelli, 2015, Santinelli et al., 2015, Santinelli et al., 2013, Santinelli, 16 2010). 17

The main scientific objective of the cruise reported here was to add knowledge to the different scales and magnitudes of variability and trends in circulation, hydrography, and biogeochemistry of the Mediterranean Sea. Key variables were measured in strategic regions in order to understand changes, the reason for occurrence, and the drivers. In this context, this cruise is part of the Med-SHIP and GO-SHIP long-term repeat cruise section that is conducted at regular intervals in the Mediterranean Sea to observe changes and impacts on physical and

- 24 biogeochemical variables.
- 25 The following science questions were addressed:

What are the long-term changes and/or trends in physics and biochemistry in the
 Mediterranean Sea, including all the sub-basins?

28 2. How is the hydrographic situation in the Mediterranean developing further after the EMT
29 and WMT? Is there still a tendency of the system to return to the pre-EMT situation and is there
30 a similar trend in the WMed?

31 3. How are eddies distributed in the EMed and WMed during the cruise? Do they differ in32 the sub basins? To what extent is heat and salt transferred into the vertical by eddies in the

1 WMed and EMed during the cruise period?

2 4. What is the uptake rate of the anthropogenic carbon in the Mediterranean and is this3 changing over time?

4 5. What is the extent of the variability and trends in the inventory of biogeochemical5 variables (including oxygen, nutrients and dissolved organic carbon)?

6 6. What are the baseline values of rarely measured Essential Ocean Variables (EOVs) such7 as dissolved organic carbon (DOC)?

8

**9 2. Data Provenance**

The survey was carried out on the German RV Maria S. MERIAN from 2nd of March to 3rd of April 2018. The cruise started on Iraklion, Greece and ended in Cadiz, Spain. The main focus of the cruise was on an east-west transect across the Western and Eastern Mediterranean Sea (figure 1) starting east of Crete and ending near the Strait of Gibraltar, which is a repeating hydrographic line in GO-SHIP (MED1). Difficulties with diplomatic authorizations for Marine Scientific Research (MSR) in the disputed EEZ between Greek and Turkey made it impossible for us to carry out measurements in this area, so that no data were obtained east of Kasos Strait.

---

## Author Response (AR2)

Dear Dr. Manzella,

We would also like to thank you for the efforts you have made to improve our manuscript and would like to apologize that we did not list the reviewers' questions and comments before the revised manuscript.

To distinguish the corrections, we have now marked the new changes based on your suggestions with grey.

**Comment 1a)** p2, l17: Referee 1 asked: 'Say somewhere in the introduction that this cruise is part of the MedShip long-term repeat cruise section that is conducted every 5 years in Mediterranean Sea to observe changes and impacts on physical and biogeochemical variables'.

*Re: The comment of reviewer 1 was taken into account and a change in the introduction was made as requested by reviewer 1 (see page 5, 21-28, marked in yellow). However, we also now changed the last sentence in the abstract (page 2, 16-18, marked in grey) exactly the same according to your objection.*

**Comment 1b)** p2, l17: 'The title, actually, is somewhat misleading, there is barely any text describing variability and no mention of trends. A more adequate title would simply be "Physical and Biogeochemical Parameters of the Mediterranean Sea during a Cruise with RV MARIA S. MERIAN in March 2018". Please consider the referee comment. '

*Re: We agree with your objection that the title is somewhat misleading and changed it according to your suggestion. This publication deals with the data and data quality and not with their interpretation.*

**Comment 2)** p2, l21: 'During the discussion period it was noted that the data are stored in;

https://cchdo.ucsd.edu/cruise/06M220180302 and https://doi.pangaea.de/10.1594/PANGAEA.905887. Please add both storage links'

*Re. The second link you mentioned, was already included (see page 5, line 5). We added the first link (page 5, line 7, marked in grey).*

**Comment 3)** p6, l1: Referee 1 underlined that a basin where convective events are occurring, the interannual variability could be of the same order or even higher than seasonal variability. Referee practically asked you to change this sentence.

*Re: To our opinion, reviewer 1 wanted to state that there are also other forces than the influence of LIW (respectively of changes of LIW due to the EMT) on the WMT like interannual variability of convective events due to atmospheric forcing. This objection was taken into account (see page 5, line 21-28, marked in yellow). The steady state hypothesis is not referring to interannual or seasonal scales but too longer periods like the climate scale, specifically the decadal time-scale of repeat hydrography. This sentence was not questioned by reviewer 1.*

**Comment 4)** p6, l6: 'When you talk about Cant, use "ant" as subscript'

*Re: We now use ant as subscript in $C_{ant}$ (page 6, 3-8, marked in grey).*

**Comment 5)** p17, l9-10: Rewrite this as "Dissolved Inorganic Carbon (DIC), pH, Total Alkalinity (TA) and carbonate ion ($CO_3^{2-}$) were measured at selected stations and depths (table 9)".

*Re: We included "were measured" (page 17, line 10, marked in grey).*

**Comment6)** p17, l13: 'Please specify if and how the inorganic has been transported. Was the PMEL procedures applied? (https://www.pmel.noaa.gov/co2/files/dic_sample_technique_revised_5-17-10.pdf)?  Probably this was one of the points on data quality raised by referee 1 ('fair data quality') '

*Re: We don't fully understand this question. All samples were measured on-board following the procedures described in the paper. We now state this in the first paragraph of section 3.8 (page 17, line 13, marked in grey).*

**Comment 7) and 8)**  p24, l24 and p 27,l13:  'Based on the question posed in the introduction and my additional comment page 3 line 17, can you briefly specify the scales resolved by data? What about trends? '

*Re: We now removed "variability and trends" in the title (Comment 1b). The questions we addressed in the introduction should only help to understand the motivation for the cruise. However, in ESSD the focus of the paper is on the data itself, their quality, where to find them, used methods, etc. but not on the scientific interpretation. To discuss scales and trends further analysis of the data has to be made. Basically, we are resolving sub-basin geographic scales at sub-decadal temporal scales – typical of the GO-SHIP repeat hydrography program. This is mentioned in the introduction of the paper.*

(All page and line numbers refer to the manuscript ESSD-2020-82-Editor.pdf which is sent together with the letter.)

With best regards

Dagmar Hainbucher